# Forecasting Company Fundamentals

**Felix Divo**                                                                                           *felix.divo@cs.tu-darmstadt.de*
*AI & ML Lab, Computer Science Department, TU Darmstadt, Germany*

**Eric Endress**                                                                                                         *endress@acatis.de*
*ACATIS Investment, Frankfurt am Main, Germany*

**Kevin Endler**                                                                                                           *endler@acatis.de*
*ACATIS Investment, Frankfurt am Main, Germany*

**Kristian Kersting**                                                                       *kersting@cs.tu-darmstadt.de*
*AI & ML Lab, Computer Science Department, TU Darmstadt, Germany*
*Hessian Center for AI (hessian.AI), Darmstadt, Germany*
*German Research Center for Artificial Intelligence (DFKI), Darmstadt, Germany*
*Centre for Cognitive Science, TU Darmstadt, Darmstadt, Germany*

**Devendra Singh Dhami**                                                                                   *d.s.dhami@tue.nl*
*Mathematics and Computer Science Departement, TU Eindhoven, Eindhoven, Netherlands*

**Reviewed on OpenReview:** *https://openreview.net/forum?id=haf78jerSt*

## Abstract

Company fundamentals are key to assessing companies' financial and overall success and stability. Forecasting them is important in multiple fields, including investing and econometrics. While statistical and contemporary machine learning methods have been applied to many time series tasks, there is a lack of comparison of these approaches on this particularly challenging data regime. To this end, we try to bridge this gap and thoroughly evaluate the theoretical properties and practical performance of 24 deterministic and probabilistic company fundamentals forecasting models on real company data. We observe that deep learning models provide superior forecasting performance to classical models, in particular when considering uncertainty estimation. To validate the findings, we compare them to human analyst expectations and find that their accuracy is comparable to the automatic forecasts. We further show how these high-quality forecasts can benefit automated stock allocation. We close by presenting possible ways of integrating domain experts to further improve performance and increase reliability.

## 1 Introduction

Company fundamentals (CFs) are a set of metrics that summarize the current financial state of a business based on aggregate statistics, like the total revenues, profit, assets, and many other indicators (Kumbure et al., 2022). They are key in many different contexts, including controlling, compliance with regulations, assessing financial stability, and—very importantly—active and passive investing. These key performance indicators (KPIs) are vital in assessing the financial well-being of companies and, in turn, determining their appeal for investing. It has been shown that they are decent proxies for a company's future performance (Wafi et al., 2015; Feng et al., 2020). Through the lens of value investing, practitioners look to invest in companies whose shares appear underpriced relative to their "intrinsic value", for which CFs are indicators. They anticipate the market eventually recognizing and correcting the undervaluation, yielding new growth in the assets invested earlier. Factor investing is stricter, where only the best assets ranked by a pre-determined static rule are chosen (Feng et al., 2020), thereby overcoming some human biases and

information overlad (Rasmussen, 2003). For example, one might select companies proportional to some CF value. A common choice would be the EV/EBIT, i.e., their enterprise value per earnings before interest and taxes (Guida & Coqueret, 2018). Similarly, analysts following different investment paradigms also pay close attention to company fundamentals as essential signals.

Specifically for a value or factor investor, knowing future company fundamentals would be enormously beneficial in making successful investment decisions measured in total share price appreciation. For example, it has been shown that if these CFs had been known prospectively, investments could have been made much more successfully (Alberg & Lipton, 2018; Chauhan et al., 2020; Downey, 2020). For instance, using a *clairvoyant model* that always "predicts" the correct value without error, Chauhan et al. are able to reach a simulated annualized return of $\sim 40\%$ (for one year of looked-ahead). This naturally leads to the question of whether reliably forecasting CFs using machine learning models can enhance investment decisions. This is particularly promising since the intersection of machine learning (ML) and quantitative financial analysis has opened new possibilities for predicting company fundamentals, a domain traditionally dominated by statistical methods and manual assessments by analysts with very specific domain expertise. When instead of the clairvoyant model, Chauhan et al. (2020) used real forecasts with inevitable prediction errors, this improved their lookahead factor model's annualized return by 3.7 percentage points over a quantitative factor model based on merely current CFs. On the other hand, the work by Downey (2020) did not show a significant benefit of using automated forecasts. This uncertain yet promising state of affairs motivates forecasting CFs as precisely and reliably as possible to serve as building blocks for factor models. Subsequently, we can rigorously assess the benefit of these prognoses to factor investing as a case study.

Specific challenges in this type of financial time series demand a thorough comparison of models. The data is very diverse, as companies across regions and industries can exhibit vastly different dynamics. Also, different variables that mirror different business processes and behave differently, yet depend on each other. For example, a company's revenue is usually a rough indicator of its income if their typical ratio is known from the last years. Furthermore, the time series are predominantly non-stationary, meaning that the distributions and dependencies of the variates change dramatically over time (cf. Section 4.1.2 and Schmitt et al. (2013)). This reaches far beyond the mere increase in the mean market capitalization of companies and accompanying heteroscedasticity (Kwon et al., 2023) since the interplay of the metrics and behavior of companies varies over time as markets develop. For example, the role of intangible and, specifically, digital assets changed dramatically over the last decades (Lim et al., 2020). Additionally, many sectors, like tourism, at least temporarily experienced significant transformations in the pandemic years starting from 2020 (cf. Section 4.1.3). Our concrete dataset of CFs contains further specific challenges, opening up many interesting research opportunities. First of all, the number of samples is rather small compared to many other deep learning datasets, considering that only 2578 companies fit our selection criterion explained in Section 4.1. Also, many forecasting models were developed for longer time series instead of quarterly data, available for less than two decades. For example, the Prophet model (Taylor & Letham, 2018) assumes "business time series", i.e., daily samples with the typical effects of weekends and holidays, to be applied most effectively. Furthermore, the selected covariates contain a lot of information while exhibiting complicated dynamics. This puts the models in danger of overfitting, especially the high-capacity ones. Furthermore, when forecasting time series in general, it is often unclear which model and configuration is the most appropriate (Subrahmanyam, 2010; Kumbure et al., 2022; Ma & Fildes, 2023). Luckily, the nature and amount of data permit a wide range of models to be used. Conversely, this necessitates a comprehensive analysis to determine the best models for each goal. These challenges make it a particularly interesting case study for research into ML forecasting.

Applying forecasting methods to CFs opens many interesting secondary research opportunities. For instance, one could investigate if latent patterns of social behavior are revealed by inspecting the forecasts and what the ML models attend to (Kelly & Xiu, 2023, Sec. 1.1). Furthermore, uncovering and quantifying correlations and interdependencies between sectors and individual companies might help reveal the structure of complicated and interwoven businesses that would otherwise remain opaque to outside observers.

**Key Contributions**  After identifying the wide research gap of a comprehensive qualitative *and* quantitative overview of company fundamentals forecasting methods, our main contributions can be summarized as the following:

(i) Effective data selection and preprocessing measures appropriate to deal with the vast range of magnitudes and the exponential nature of the variables under consideration.

(ii) A balanced examination of classical statistical and contemporary deep learning forecasting models. We summarize their construction and classify them according to their capabilities and properties.

(iii) An investigation of the model's fitness for forecasting company fundamentals, a comparison to human forecasts, the availability of uncertainty estimates, and their applicability to investment strategies by factor model backtesting.

(iv) A discussion of applicable methods for interpretability, explanations, and incorporation of expert knowledge.

**Overview**  We start by reviewing related research to contextualize this work and further motivate its necessity (Section 2). We then describe the models under investigation and highlight their specific strengths and weaknesses (Section 3). Finally, we empirically compare the models on a real-world dataset of company fundamentals and perform an in-depth comparison of the models based on that (Section 4). We continue by discussing interpretability and opportunities for expert involvement (Section 5), and finally conclude with an outlook on possible future research directions (Section 6).

## 2   Related Work

**Use of Past Company Fundamentals**  For a long time, CFs have been widely used in many economic contexts by both practitioners and researchers (Hayes, 1961, pp. 155ff). They serve as a foundation to assess the current success and financial sustainability of businesses. As objective and quantitative measures, they are often beneficially employed in factor investing (Aspris et al., 2013; Wafi et al., 2015; Muhammad & Ali, 2018; Guerard et al., 2015; Song & Lee, 2019). However, basing investment decisions solely on fundamental values is viewed as naive and risky by many (Davis, 2017; Kok et al., 2017; Lev & Srivastava, 2022). For example, the now famous short squeeze of the *GameStop Corp.* stock in January 2021 (Umar et al., 2021) showed that additional signals like news and social media sentiment should be used to complement CFs for trading descisions (Chen et al., 2020). However, this specific stock was overall unattractive to (value) investing back then. These efforts complement the work in this research, where we primarily establish effective forecasts of CFs as building blocks for further research. For instance, many stock price prediction methods rely on a multitude of factors, including past CFs (Kumbure et al., 2022; Rasekhschaffe & Jones, 2019; Ta et al., 2020; Guida & Coqueret, 2018), and could leverage CF forecasts in the future.

**Forecasting Company Fundamentals**  There have been some prior attempts at forecasting fundamentals. Most notably, experiments with an idealized clairvoyant model always predicting the known future and, subsequently, imperfect LSTMs showed that lookahead factor models (LFMs) based on CF forecasts can improve returns of portfolios (Alberg & Lipton, 2018; Chauhan et al., 2020). Sun (2019) studied fundamentals forecasts with a few simple deep learning (DL) models, yet did not apply them to stock selection. Following the work of Alberg & Lipton (2018), Downey (2020) first replicated the "crystal ball" portfolio based on perfect fundamental forecasts and confirmed its impressive hypothetical performance for stock allocation. However, none of the investigated models (Random Forests, Gradient Boosting, Support Vector Machines, Multilayer Perceptron, Linear Regression) could forecast CFs with sufficient reliability to affect investing positively. Guerard et al. (2015) used analysts' expectations on companies' revenue and found this to be a strongly beneficial signal for stock selection. This again highlights the need for accurate company fundamental forecasts.

Table 1: Overview of the models considered in this evaluation. It shows if they are local, global, global deep learning (DL), or global pretrained (PT) models. Further properties are if they are autoregressive (AR); the number of predicted variates; their use of additional covariates, static variables, or RevIN; and their ability to provide probabilistic forecasts.

| Model | Reference | Type | AR | Capabilities | | | | |
|-------|-----------|------|-----|-------|--------|---------|-------|-------|
| | | | | # Var. | Covar. | Statics | RevIN | Prob. |
| Mean Value | Herzen et al. (2022) | local | no | uni | ✗ | ✗ | ✗ | ✗ |
| ARMean($p$) | Herzen et al. (2022) | local | yes | uni | ✗ | ✗ | ✗ | ✗ |
| ARMA($p, q$) | Box & Jenkins (1976) | local | yes | uni | ✗ | ✗ | ✗ | ✓ |
| ARIMA($p, d, q$) | Box & Jenkins (1976) | local | yes | uni | ✗ | ✗ | ✗ | ✓ |
| VARIMA($p, d, q$) | Herzen et al. (2022) | local | yes | multi | ✓ | ✗ | ✗ | ✓ |
| AutoARIMA | Hyndman & Khandakar (2008) | local | yes | uni | ✗ | ✗ | ✗ | ✓ |
| AutoTheta | Hyndman & Billah (2003) | local | yes | uni | ✗ | ✗ | ✗ | ✓ |
| Prophet | Taylor & Letham (2018) | local | yes | uni | ✗ | ✗ | ✗ | ✓ |
| Linear Reg. | Herzen et al. (2022) | global | no | multi | ✓ | ✓ | ✗ | ✓ |
| Random Forest | Breiman (2001) | global | no | multi | ✓ | ✓ | ✗ | ✗ |
| DLinear & NLinear | Zeng et al. (2023) | DL | no | uni | ✓ | ✓ | ✓ | ✓ |
| RNN (LSTM) | Hochreiter & Schmidhuber (1997) | DL | yes | multi | ✗ | ✗ | ✗ | ✓ |
| RNN (GRU) | Cho et al. (2014) | DL | yes | multi | ✗ | ✗ | ✗ | ✓ |
| Block RNN | Herzen et al. (2022) | DL | no | multi | ✗ | ✗ | ✗ | ✓ |
| TCN | Bai et al. (2018) | DL | no | multi | ✓ | ✗ | ✓ | ✓ |
| Transformer | Vaswani et al. (2017) | DL | yes | multi | ✓ | ✗ | ✓ | ✓ |
| TFT | Lim et al. (2021) | DL | no | multi | ✓ | ✓ | ✓ | ✓ |
| N-BEATS | Oreshkin et al. (2020) | DL | no | multi | ✓ | ✗ | ✓ | ✓ |
| N-HiTS | Challu et al. (2023) | DL | no | multi | ✓ | ✗ | ✓ | ✓ |
| TiDE | Das et al. (2023) | DL | no | multi | ✓ | ✓ | ✓ | ✓ |
| xLSTM-Mixer | Kraus et al. (2024a) | DL | no | multi | ✓ | ✗ | ✓ | ✓ |
| Chronos | Ansari et al. (2024) | PT | yes | uni | ✗ | ✗ | ✗ | ✓ |

**Financial Machine Learning** There is a plethora of surveys on the broad topic of financial machine learning (Hoang & Wiegratz, 2023; Nazareth & Ramana Reddy, 2023; Mosavi et al., 2020). Most of them specialize in price and movement prediction of stocks, forex, or similar assets (Sezer et al., 2020; Nabipour et al., 2020; Kumbure et al., 2022; Htun et al., 2023; Zhang et al., 2023; Kelly & Xiu, 2023). Existing surveys often do not compare different approaches empirically, or if so, not on the same data and thereby do not offer reliably comparable quantitative results (Spiliotis, 2023; Mahmoud & Mohammed, 2021). There is very little comparative research into the focus topic of this work: CF forecasting.

## 3 Model Selection for Forecasting

We selected various models to provide a thorough overview of the available modeling toolbox, as summarized in Table 1. Univariate forecasting models assume all variates to be independent, thereby effectively modeling them as separate time series. Regarding notation, we will denote a univariate time series of length $T$ as a sequence $x_1, \ldots, x_T \in \mathbb{R}$, and use bold $\boldsymbol{x}_1, \ldots, \boldsymbol{x}_T \in \mathbb{R}^D$ for multivariate time series. The goal is to predict the $h \in \{1, \ldots, H\}$ steps ahead by looking back at the last $B \leq T$ time steps. Some models directly predict $H > 1$ future steps, whereas the other models forecast one step at a time in an autoregressive fashion (see $AR$ in Table 1). This is sometimes called single-step and multi-step forecasting, respectively (Lim et al., 2021). In addition to target variates and covariates, some models can additionally be conditioned on static metadata per company, such as its primary business sector.

In addition to the preprocessing steps described later in Section 4.1.2, we evaluated applying Reversible Instance Norm (RevIN) (Kim et al., 2022) in all non-recurrent deep learning models as shown in Table 1. It removes some nonstationarity and possible shifts in distribution. RevIN removes the mean and rescales to

unit variance along each individual time series and variate $x$. Furthermore, the result is then transformed by an affine transform with learnable $\gamma$ and $\beta$:

$$\text{RevIN}(x_t) = \gamma \left( \frac{x_t - \mathbb{E}\left[x\right]}{\sqrt{\text{Var}\left[x\right] + \epsilon}} \right) + \beta.$$

We found it to be mostly beneficial, as shown quantitatively in detail in Appendix A.4, and will thus apply it throughout the paper.

### 3.1 Local Models

This class of models is estimated on a single time series, effectively modeling all companies as entirely separate processes. While fitting a single model is comparatively fast, estimating a separate model per company is usually much more resource-intensive than learning a shared global model. We consider the following local models for evaluation, also containing some simplistic baselines like Mean Value and ARMean($p$).

**Mean Value**  This simple baseline model repeatedly predicts the mean value of each variate it has seen during training: $\boldsymbol{x}_{T+h} = \frac{1}{T} \sum_{i=1}^{T} \boldsymbol{x}_i$. It is included as the arguably simplest model one should consider.

**Autoregressive Mean (ARMean($p$))**  As another simple baseline, this predicts a simple arithmetic mean of the last $p$ values as

$$x_t = \frac{1}{p} \sum_{i=1}^{p} x_{t-i}.$$

**Autoregressive Moving Average (ARMA($p, q$))**  This univariate model assumes the time series to be recursively defined by an autoregressive (AR) and moving average (MA) component as

$$x_t = \mathbb{E}[x] + \sum_{i=1}^{p} \phi_i x_{t-i} \sum_{i=1}^{q} \psi_i \epsilon_{t-i},$$

with noise terms $\epsilon_t$ and the expected value $\mathbb{E}[\cdot]$ of the time series (Box & Jenkins, 1976). The parameters $\phi_i$ and $\psi_i$ are shared across the entire duration. If the time series has zero mean, the special case where $\phi_i = \frac{1}{p}$ is equivalent to an ARMean($p$) model.

**Autoregressive Integrated Moving Average (ARIMA($p, d, q$))**  The ARIMA($p, d, q$) model improves upon ARMA($p, q$) by first computing $d$ differencing steps $x'_t = x_t - x_{t-1}$ and using that as the basic time series to model (Box & Jenkins, 1976). VARIMA($p, d, q$) is a vector-based variant of ARIMA($p, d, q$) and would thus be a good fit to model dependencies between multiple variates (Herzen et al., 2022). It is equivalent to a VARMA($p, q$) model (Lütkepohl, 2006) applied after $d$ differencing steps.

**AutoARIMA**  In practice, it is often unclear how $p$, $d$, and $q$ shall be chosen. AutoARIMA introduces an automated algorithm for determining them in a data-driven search (Hyndman & Khandakar, 2008).

**Theta Model (AutoTheta)**  This model was first described by Assimakopoulos & Nikolopoulos (2000) but was later simplifed siginificantly (Hyndman & Billah, 2003). It first models a surrogate time series $y_{t,\theta}$, which is connected to $x_t$ through its second-order derivative: $y''_{t,\theta} = \theta x''_t$. This can then be used to derive $y_{t,\theta} = a_\theta + b_\theta(t-1) + \theta x_t$, with $a_\theta$ and $b_\theta$ derived from the data. The final forecast is then derived from the cases $\theta = 0$ and $\theta = 2$. AutoTheta then automatically optimizes its parameters (Fiorucci et al., 2016).

**Prophet**  This model was initially geared toward business time series and conceptually decomposes the observations into trend, seasonality, holidays, and noise components, with separate univariate models each (Taylor & Letham, 2018). Since we model quarterly and international data, we omit the seasonality and holiday components in this work. We found that the model performance increased when normalizing per time series instead of across them, similar to RevIN, and thus applied it in all evaluations of the Prophet model (cf. Appendix A.3).

### 3.2 Global Models

This class of models aims to learn common representations of the time series of all companies at once to capture the dynamics shared among them. They are also usually much faster to train since only a single model needs to be estimated instead of one per company. We consider the following global models for evaluation.

**Linear Regression** This autoregressively performs a multiple and multivariate ordinary least squares (OLS) linear regression for each predicted value on the last $n$ lags of each of the target variates and context covariates.

**Random Forest** This is an ensemble of multiple regression trees, where each one regresses on the same observations as for linear regression (Breiman, 2001). In particular, separate random subsets of the training data are used to grow multiple trees in the forest. Separate forests are learned for each predicted lag.

**DLinear & NLinear** This pair of models was proposed in 2023 as a simple baseline to compete with comparatively intricate Transformer-based models in long-term time series forecasting (Zeng et al., 2023). Both are only linear regression models, where multiple time steps are forecasted at once based on multiple past steps. Predictions for different variates share weights. DLinear acts on a decomposition into trend and residual, where NLinear only normalizes by subtracting the last value. While being rather shallow models, they are grouped with the other deep learning (DL) models in Table 1 since they are trained similarly with gradient descent.

**Recurrent Neural Network (RNN, specifically LSTM and GRU)** While not being the first ever RNNs, Long Short-term Memory models (LSTMs) (Hochreiter & Schmidhuber, 1997) popularized the concept of iteratively consuming an input sequence to produce both an output and a hidden state propagated to the next step. Using several gating functions, very deep LSTM networks can be trained via backpropagation through time. The recurrent models used in this work are equivalent to the DeepAR model of Salinas et al. (2020). Gated recurrent units (GRUs) simplify LSTMs, often reaching similar performance with fewer parameters (Cho et al., 2014).

**Block RNN (LSTM and GRU)** This is a variant of the standard RNN introduced above, where the time series is modeled as "blocks" of several observations passed to the network as one. This is akin to RNNs trained on multivariate data.

**Temporal Convolutional Network (TCN)** This model applies the basic building blocks of convolutions and residual connections to time series (Bai et al., 2018). They are crafted to induce a bias appropriate for time series. Firstly, the convolutions are causal, meaning the kernel only receives data from preceding timesteps. Secondly, they are dilated, allowing the model to look further back than in a classical convolutional neural network with the same kernel size. Finally, skip connections are employed to stabilize training with 1x1 convolutions where channel dimensions misalign.

**Transformer** Originally proposed as a model for natural language (Vaswani et al., 2017), Transformers have since been used as general sequence-to-sequence models. We model the data as a sequence of tokens, i.e., vectors with added position information. Then, for each individual token, we add a combination of projections of other tokens proportional to Scaled Dot-Product Self-Attention: $\text{softmax}\left(QK^T/\sqrt{d_k}\right)V$. Here, the query $Q$, key $K$, and value $V$ are learnable linear projections of the respective input tokens, and $d_k$ is the token dimension. Then, skip connections and simple feed-forward layers are used to transform the tokens. After several such layers, the prediction is extracted. Note that we use the specific implementation from darts (Herzen et al., 2022).

**Temporal Fusion Transformer (TFT)** Recurrent neural networks, as in LSTMs, and attention-based architectures of the Transformer family of models are often viewed as alternative and separate approaches. Temporal Fusion Transformer is a take on combining them to benefit from both (Lim et al., 2021). To this end, an LSTM encoder-decoder architecture is used for short-term patterns, where the outputs are subsequently fed into a Transformer-like layer for long-term predictions. The architecture employs various gating mechanisms to learn which variables to use, when to propagate features from the LSTM outputs, and how to propagate features inside the Transformer-like layer.

**N-BEATS** It is often beneficial to decompose challenging tasks into a sequence of easier ones. N-BEATS, therefore, comprises a stack of doubly residually connected blocks, where each block performs both a forecast and a backcast (Oreshkin et al., 2020). The backcast is subtracted from the input and only then passed to the subsequent block, in effect removing the component used by the block and only leaving the following block with the residual. Similarly, the complete forecast is computed as the sum of individual forecasts, effectively modeling it as an additive decomposition. Each back- and forecast is computed based on predicting basis function coefficients to ease training and allow for inductive biases. We use the multivariate extension of Darts (Herzen et al., 2022).

**N-HiTS** This model is an extension of N-BEATS, particularly made for long sequences, where hierarchical input and outputs are used in addition to the doubly residual stacking (Challu et al., 2023). This model assumes that the signal consists of separate low- and high-frequency components. Consequently, the blocks in the stacks are arranged to progress from processing and generating at low to high resolution.

**Time-Series Dense Encoder (TiDE)** This model consists of an encoder-decoder architecture: It consumes the lookback window, static metadata, and covariates; then produces a low-dimensional embedding; and finally decodes it to the forecast (Das et al., 2023). To stabilize training, residual connections are added within the individual encoder and decoder layers and around the entire architecture. Additionally, the covariates are first projected, and the final forecast is un-projected per time step with another residual.

**xLSTM-Mixer** Recently, mixing emerged as a new paradigm for deep learning architectures, where different dimensions of the data are projected with weight sharing in an alternating fashion (Tolstikhin et al., 2021). This has since been successfully transferred to time series: xLSTM-Mixer (Kraus et al., 2024a) combines recurrent xLSTM modules (Beck et al., 2024) with three different mixing stages to perform forecasting. Namely, the time dimensions are first projected per variate, then time and variate data are jointly projected in the recurrent step, and a final view mixing reconciles a forward and backward view of the series into the output forecast.

**Chronos** In language modeling, self-supervised pretraining on vast amounts of data has proven tremendously effective at learning the basic structure of texts (Zhao et al., 2024). This paradigm has recently been transferred to deep time series forecasting (Kraus et al., 2024b). Chronos is one such model closely following the structure of typical transformer-based language models (Ansari et al., 2024). The method is trained on large amounts of real and synthetic time series normalized by mean scaling and discretized like text tokens. The model is both pretrained and deployed by autoregressive discrete next-token prediction. We use the improved variants Bolt Small and Bolt Base.

## 4 Evaluation

To assess the models' applicability to the challenging CF forecasting task, we empirically evaluate them in several settings and across many different time scales. We continue to apply the forecasts to automatic stock selection to determine the benefit in that field of application.

### 4.1 Forecasting Performance

Good data quality is crucial for accurate forecasting. Therefore, significant effort was put into curating the covariates and targets to predict and cleaning the data to provide a consistent database. This later allows the models to capture the common temporal dynamics of company fundamentals across different companies.

#### 4.1.1 Selected Indicators

Based on reliable availability and the experience of experts, 20 indicators were deemed the most relevant for predicting the future fundamentals of a company. All were used as context to predict five of them as target variables. On the one hand, restricting the forecast to only a subset of all indicators deemed the most insightful improved the performance slightly. Yet, learning a model forecasting all remaining five targets jointly is a beneficial multi-task setting helping against overfitting on this rather small dataset (Zhang & Yang, 2022). Indicators derived from cash flow and income statements are based on intervals instead of reflecting the financial state at a single point in time, as balance sheets do. Therefore, we consider their average over the last twelve months, denoted as *LTM*. The selected covariates used only to provide context to the model are: Cost Of Revenues (LTM), Total Other Operating Expenses (LTM), Capital Expenditure (LTM), Income Tax Expense (LTM), Total Interest Expense (LTM), Levered Free Cash Flow (LTM), Cash from Financing (LTM), Cash from Investing (LTM), Total Cash and Short-Term Investments, Total Current Assets, Total Assets, Total Current Liabilities, Total Liabilities, Total Debt, and Sector Revenue Share (LTM). Based on expert experience, we chose the following five target variables as the most important ones for decision-making: Total Revenues (LTM), Operating Income (LTM), Net Income (LTM), Cash from Operations (LTM), and Total Equity.

We experimented with providing the model with additional static context for each company. Namely, we added region information (Americas/Europe/Japan/Rest of Asia) and sector classifications according to the Global Industry Classification Standard (GICS) by MSCI and S&P. We encoded them as numeric one-hot statics for the global models that can use it (see Table 1). However, we only retained the latter since the region information did not noticeably change the forecasts. This is likely due to it being very coarse and of limited relevance in globally diversified companies. See Appendix A.4 for details.

#### 4.1.2 Data Preparation

The primary quarterly company fundamentals data was obtained by integrating several proprietary data sources from S&P Global. Considerable manual preprocessing was required to bring information from balance sheets, income statements, and cash flow statements into a single format. In particular, restatements and pro forma/hypothetical statements introduced significant inconsistencies and jumps in the data if not accounted for appropriately. All values were converted into Euros for the sake of comparability. Companies were included if (1) they were publicly traded throughout the entire period and, therefore, published reports, and (2) they had at least 1 billion Euros in market capitalization at some point in the time span. Smaller businesses were excluded because they typically show different dynamics than larger ones. Additional highly atypical companies were removed if they contained more than one data point in any feature that was at least 30 standard deviations from the mean after the normalization described below. We found two rounds of such cleaning of extreme outliers in the long tails (cf. Appendix A.2) to be sufficient, which removed only 19 companies. This left 2527 companies in the period from 2009 Q1 to 2023 Q3.

Appropriate data representation and normalization are critical in machine learning applications. In the particular case of CFs, one has to counteract the different orders of magnitude of the indicators to ensure more similar numerical ranges of the data. We, therefore, propose an effective domain-specific normalization. For each company individually, all variates derived from the income statement are normalized by Total Revenues, and all variates from the balance sheet are normalized by Total Assets. Total Revenues and Total Assets are divided by each other, respectively. This proved to be more effective than a generic Yeo–Johnson power transformation (Yeo & Johnson, 2000). Finally, the data was normalized to have zero mean and unit variance per feature over all time steps and companies. Details on the resulting challenging dataset, including its moments, high degree of non-stationarity, and diverse seasonality, are provided in Appendix A.2.

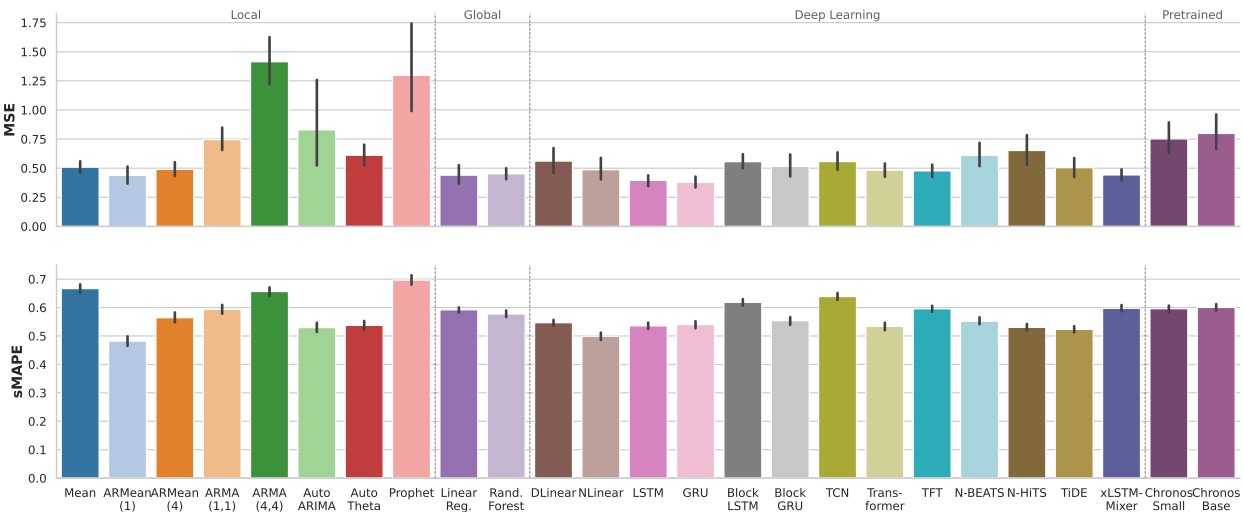

Figure 1: **Global and deep models generally provide better deterministic forecasts than local and pretrained models measured by MSE, but are much more complex.** Lower is better. It shows the mean and standard deviation over all five target features, forecast horizon steps, and folds.

### 4.1.3 Results

We evaluated the previously described models with simulated historical forecasts, where we trained on past *in-sample* (IS) data and evaluated them on unseen *out-of-sample* (OOS) time steps. This demonstrates whether models can generalize to future unseen data—the crucial property in time series forecasting. To obtain the most realistic benchmark scenarios, we used 2009 Q1 as a fixed origin and continuously expanded the training data from only four years (16 quarters) at the beginning to 13.75 years (55 quarters) at the end. We evaluated several configurations to determine the optimal lookback window. Three years (twelve quarters) was optimal for forecasting one year (four quarters). We evaluated all models on all companies and metrics for each of the 40 simulated historical forecast chunks, each consisting of IS look-back data, IS training forecasts, and OOS testing forecasts. To search hyperparameters and validate the deep learning models' training success, we created a validation split by excluding 10% of the companies from training (but not from testing). Note that the deep learning models have thus seen less of the training data, effectively including a small generalization test to unseen companies in their evaluation. Model configurations and hyperparameters are provided in Appendix A.3. The global models with and without RevIN are compared in Appendix A.4. Due to numerical instabilities, we were not able to reliably fit ARIMA$(4, 1, 4)$ or VARIMA$(4, 0, 4)$ models, for which reason they are excluded from the following analysis.

**Deterministic Forecasts** Due to the data normalization applied previously, the scales of different features are more comparable than in the original data. However, the distribution of values is highly skewed, and we want to produce forecasts that are correct relative to the specific scale of the predicted data. We, therefore, mainly compare the deterministic predictions using the symmetric Mean Absolute Percentage Error (sMAPE) and the training criterion Mean Squared Error (MSE):

$$\text{sMAPE}(\boldsymbol{y}, \hat{\boldsymbol{y}}) = \frac{2}{H} \sum_{i=1}^{H} \frac{|y_i - \hat{y}_i|}{\max(|y_i| + |\hat{y}_i|, \epsilon)} \qquad \text{and} \qquad \text{MSE}(\boldsymbol{y}, \hat{\boldsymbol{y}}) = \sum_{i=1}^{H} (y_i - \hat{y}_i)^2,$$

where $\boldsymbol{y}$ are the targets, $\hat{\boldsymbol{y}}$ the predictions, and $\epsilon$ a small constant added for numerical stability. In addition, we provide results for Mean Absolute Error (MAE), Root Mean Squared Error (RMSE), Mean Absolute Percentage Error (MAPE), Root Squared Error (RSE), and the coefficient of determination ($R^2$). The probabilistic nCRPS (reduced to MAE for deterministic models) will be introduced later.

Table 2: **Recurrent models provide the overall best performance.** Comparison of average performance and standard deviation of the models measured by different metrics. The average and standard deviation is taken over all five target features, forecast horizon steps, and folds. The best is marked as **bold** and the runner-ups as underlined. See also Figure 2 for nCRPS/MAE.

| Models | MAE ($\downarrow$) | MSE ($\downarrow$) | RMSE ($\downarrow$) | MAPE ($\downarrow$) | RSE ($\downarrow$) | sMAPE ($\downarrow$) | R$^2$ ($\uparrow$) | nCRPS ($\downarrow$) |
|---|---|---|---|---|---|---|---|---|
| Mean | 0.239±0.03 | 0.508±0.17 | 0.705±0.11 | 3.470±1.37 | 0.507±0.09 | 0.667±0.05 | 0.493±0.09 | 0.239±0.03 |
| ARMean(1) | **0.166±0.04** | 0.438±0.24 | 0.642±0.16 | 2.299±0.80 | 0.430±0.19 | 0.482±0.06 | 0.570±0.19 | 0.166±0.04 |
| ARMean(4) | 0.198±0.04 | 0.490±0.21 | 0.685±0.15 | 2.689±0.83 | 0.484±0.15 | 0.564±0.06 | 0.516±0.15 | 0.198±0.04 |
| ARMA(1,1) | 0.234±0.04 | 0.746±0.32 | 0.847±0.17 | 3.349±1.08 | 0.752±0.27 | 0.593±0.05 | 0.248±0.27 | 0.200±0.04 |
| ARMA(4,4) | 0.294±0.06 | 1.414±0.65 | 1.163±0.25 | 4.068±1.30 | 1.466±0.67 | 0.656±0.05 | -0.466±0.67 | 0.273±0.05 |
| AutoARIMA | 0.183±0.05 | 0.829±1.21 | 0.808±0.42 | **2.214±0.89** | 0.813±1.15 | 0.529±0.05 | 0.187±1.15 | 0.149±0.04 |
| AutoTheta | 0.202±0.04 | 0.611±0.31 | 0.761±0.18 | 2.821±0.93 | 0.609±0.26 | 0.537±0.06 | 0.391±0.26 | 0.196±0.04 |
| Prophet | 0.298±0.05 | 1.297±1.21 | 1.077±0.37 | 4.190±1.33 | 1.410±1.68 | 0.697±0.06 | -0.410±1.68 | 0.265±0.05 |
| Linear Reg. | 0.195±0.02 | 0.440±0.25 | 0.644±0.16 | 3.143±1.22 | 0.432±0.19 | 0.592±0.03 | 0.568±0.19 | 0.401±0.02 |
| Random Forest | 0.196±0.03 | 0.452±0.16 | 0.663±0.11 | 2.676±1.23 | 0.449±0.10 | 0.577±0.04 | 0.551±0.10 | 0.196±0.03 |
| DLinear | 0.186±0.03 | 0.561±0.35 | 0.719±0.21 | 2.600±0.99 | 0.560±0.31 | 0.547±0.06 | 0.440±0.31 | 0.131±0.02 |
| NLinear | 0.168±0.03 | 0.487±0.29 | 0.673±0.19 | 2.406±1.03 | 0.482±0.25 | 0.499±0.04 | 0.518±0.25 | 0.135±0.02 |
| LSTM | 0.176±0.03 | 0.395±0.16 | 0.616±0.13 | 2.455±0.94 | 0.385±0.09 | 0.535±0.04 | 0.615±0.09 | **0.124±0.02** |
| GRU | 0.175±0.03 | **0.378±0.16** | 0.602±0.13 | 2.369±0.93 | **0.368±0.09** | 0.540±0.04 | **0.632±0.09** | **0.124±0.02** |
| Block LSTM | 0.218±0.03 | 0.556±0.20 | 0.733±0.14 | 3.144±1.39 | 0.573±0.24 | 0.618±0.04 | 0.427±0.24 | 0.144±0.02 |
| Block GRU | 0.189±0.03 | 0.515±0.30 | 0.695±0.18 | 2.702±1.03 | 0.522±0.31 | 0.554±0.04 | 0.478±0.31 | 0.127±0.02 |
| TCN | 0.224±0.03 | 0.557±0.25 | 0.731±0.15 | 3.302±1.44 | 0.557±0.20 | 0.639±0.04 | 0.443±0.20 | 0.155±0.03 |
| Transformer | 0.183±0.03 | 0.483±0.19 | 0.682±0.14 | 2.494±0.92 | 0.487±0.18 | 0.534±0.04 | 0.513±0.18 | 0.132±0.02 |
| TFT | 0.206±0.02 | 0.477±0.17 | 0.679±0.13 | 2.918±1.03 | 0.480±0.15 | 0.595±0.04 | 0.520±0.15 | 0.134±0.02 |
| N-BEATS | 0.192±0.03 | 0.611±0.34 | 0.757±0.19 | 2.697±1.05 | 0.615±0.31 | 0.552±0.04 | 0.385±0.31 | 0.134±0.02 |
| N-HiTS | 0.185±0.03 | 0.652±0.42 | 0.773±0.24 | 2.490±0.73 | 0.636±0.35 | 0.530±0.04 | 0.364±0.35 | 0.130±0.02 |
| TiDE | 0.179±0.03 | 0.504±0.27 | 0.689±0.17 | 2.547±0.97 | 0.498±0.21 | 0.523±0.04 | 0.502±0.21 | 0.126±0.02 |
| xLSTM-Mixer | 0.208±0.63 | 0.442±22.71 | **0.233±0.62** | 3.032±277 | 8167.136 | 0.422±0.43 | -29758.268 | 0.146±0.50 |
| Chronos Small | 0.225±0.84 | 0.751±63 | 0.264±0.83 | 3.422±344 | 8673.922 | **0.421±0.44** | -30063.893 | 0.418±0.76 |
| Chronos Base | 0.231±0.86 | 0.798±57 | 0.273±0.85 | 3.403±288 | 9316.606 | 0.423±0.44 | -30118.321 | 0.418±0.76 |

The comparison of the models is shown visually in Figure 1. In both metrics, the models fare notably differently. Overall, global models tend to give better results in terms of MSE, while the picture is more varied if performance is measured in sMAPE. Specifically, the univariate local models ARMA(4, 4) and Prophet fail to provide consistent forecasts, as their high MSE and considerable variance hint at some major mispredictions. The pretrained Chronos models fail even more drastically, possibly due to the unusually short length of the time series. While in terms of MSE, the two deep learning RNN models (LSTM and GRU) give the best results, the simplistic local model ARMean(1)— somewhat surprisingly—is best in terms of sMAPE. This highlights the need for forecasts that are not only accurate but also reliable. Therefore, we will now go over to probabilistic forecasts.

**Probabilistic Forecasts** To assess the reliability of the forecasters, we evaluated their ability to quantify uncertainty correctly. Not all models can estimate uncertainty, as shown in Table 1. Sampling forecasts can be achieved in different ways. For instance, one can introduce noise in the state space as in the ARMA, ARIMA, and AutoARIMA models or in the trend component of the Prophet model, which is similar to Monte Carlo dropout in deep learning models (Gal & Ghahramani, 2016). The remaining models can directly estimate the parameters of probability distributions. Therefore, and for a fair comparison, all remaining models perform quantile regression, where the boundaries of certain quantiles were regressed on. Note that these models are trained separately from the ones in the deterministic setting, even though represented jointly in Table 2. We have drawn 100 samples from each probabilistic model and evaluated the resulting empirical distribution against the known test data using the continuous ranked probability score (CRPS) (Gneiting & Raftery, 2007). Intuitively, we first define a helper cumulative distribution function $\mathbf{1}_{\{x \geq y\}}$ over $x$ that steps from zero to one at the scalar value $y$ that we consider the ground truth. The CRPS then measures the integrated squared difference between this helper and the true distribution $E(x)$. It is defined as

$$\text{CRPS}(E, y) = \int_{-\infty}^{+\infty} \left( E(x) - \mathbf{1}_{\{x \geq y\}} \right)^2 \, dx = \mathbb{E}_{\hat{y} \sim E} \left[ |\hat{y} - y| \right] - \frac{1}{2} \mathbb{E}_{y_1, y_2 \sim E} \left[ |y_1 - y_2| \right],$$

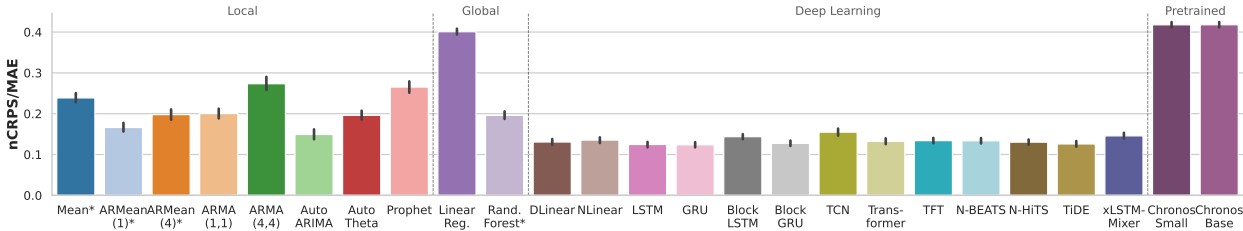

Figure 2: **Models based on deep learning are clearly superior to all other families.** The error of different models when forecasting the normalized features probabilistically. Lower is better. It shows the mean and standard deviation over all five target features, forecast horizon steps, and folds. Since the nCRPS score reduces to the absolute error (MAE) in the case of deterministic forecasts (i.e., a single sample), we include such models for direct comparison marked with *. Full results are in Table 2.

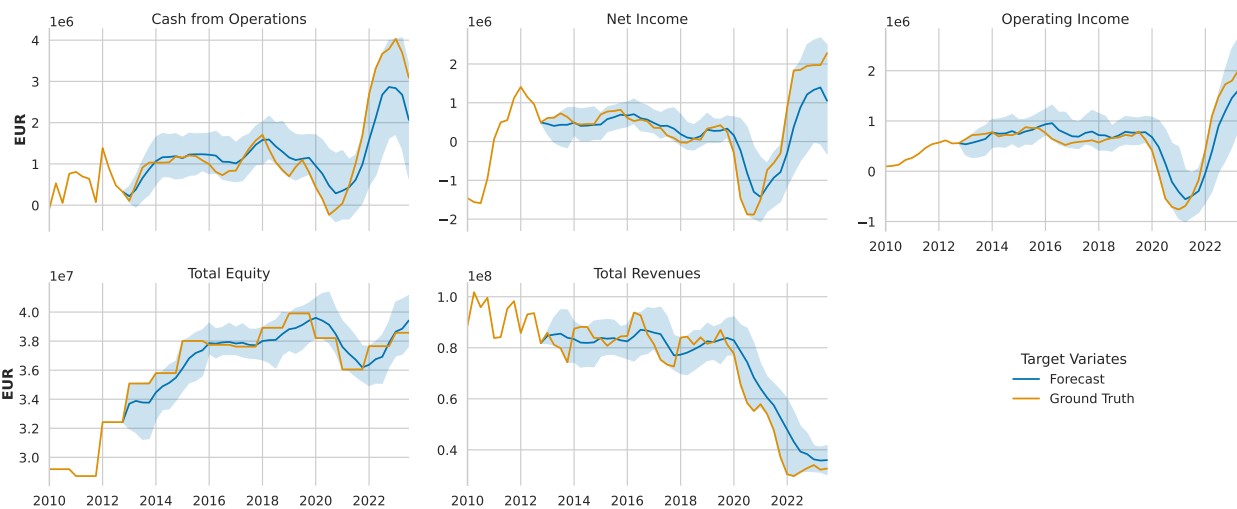

Figure 3: **Qualitative example of forecasting the fundamentals** for *Pacira BioSciences, Inc.* over several different horizons using the RNN (GRU) model. The shaded area indicates the 68% percentile.

where $E$ is the empirical forecast distribution (i.e., the samples), $y$ is the ground truth target, and $\mathbf{1}_{\{x \geq y\}}$ is an indicator that is 1 if the condition is met and 0 otherwise. This univariate metric is then averaged over all time steps and features. We choose to present the negation of that score denoted with "nCRPS" to make it consistent with the earlier metrics, where lower is better.

The results are shown in Figure 2 and Table 2. We can clearly see that all deep learning models fare much better than all others in estimating uncertainty, even though they did not see 10% of the companies before testing. This shows their strength of learning across the time series of many companies, thereby enabling good generalization and a more calibrated variance. The best models are again the RNN variants LSTM and GRU, closely followed by TiDE.

An example forecast transformed back to the original feature domain is provided in Figure 3. The normalized forecasts were inverse-transformed back into the original domain using the last-known training data to avoid any information leakage. The model successfully captures the overall dynamics of the variates while tending to lag slightly behind. In particular, the variance of the model appropriately captures the variability of the ground-truth time series.

**Discovering Patterns in Predictability** As shown in Figure 4, not all features are equally predictable by all models. For instance, the *Total Equity* is easily forecasted by ARMean(1), yet much harder for others. This is plausible, given that it only moderately changes over time and thus might cause more overfitting.

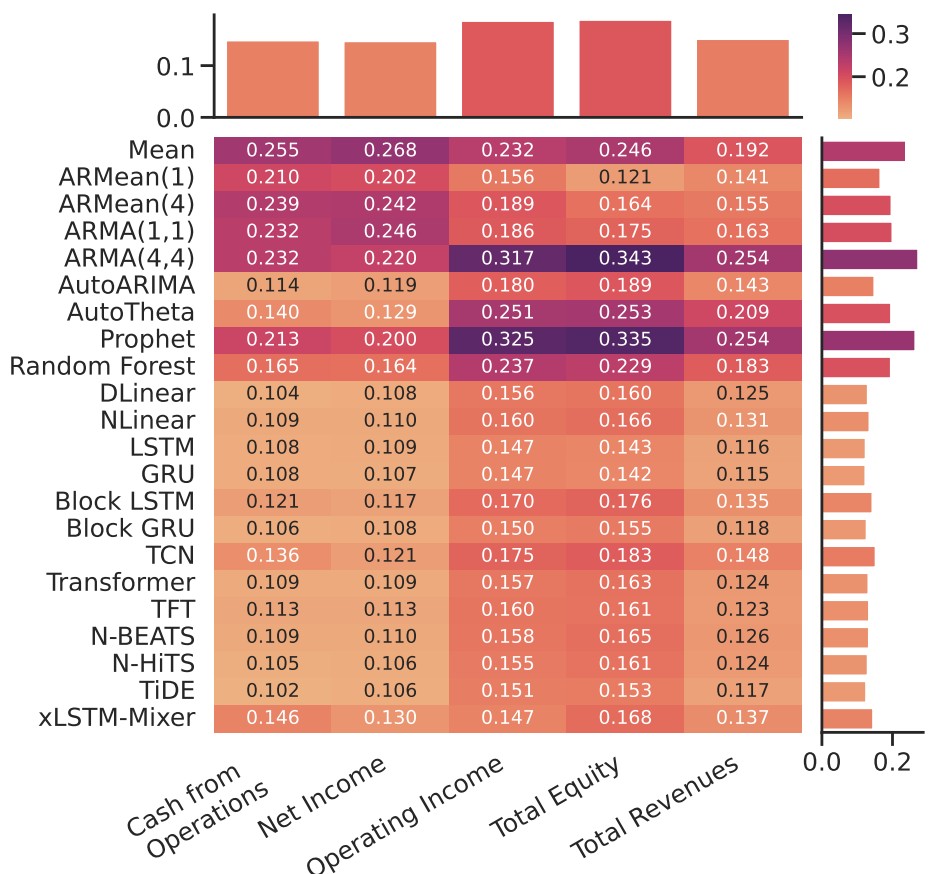

Figure 4: **Models differ substantially in the features they can successfully forecast.** This plot shows the mean nCRPS/MAE per model and feature over all folds. Lower is better. Linear Regression and Chronos, with their large errors, are excluded to make differences more discernible. The *Operating Income* and the *Total Equity* pose the greatest challenges to the models under investigation.

On the other hand, it is surprising that *Total Revenues* is relatively straightforward to predict for the DL models, indicating that some other covariates are predictive of it. The fact that the *Operating Income* is more challenging suggests that the cost of revenues varies significantly, affecting the variate in unpredictable ways. Consequently, the *Net Income* is affected, albeit reduced, possibly due to steadier taxation and interests. By cumulating it over time, one derives the, therefore, equally predictable *Cash from Operations*.

Global Black Swan events like the COVID-19 pandemic starkly impact forecasting performance, as shown in Figure 5. Note that for most variates and models, this extends far beyond the difficulties in forecasting in the respective periods, like here, 2020 Q1 and onward. On the one hand, starting with 2019 Q2, the forecast evaluation is already infected with COVID-19 effects since we forecast one year into the future. Furthermore, the abnormal data from the pandemic period is then used to learn forecasts for later times, which no longer share the same pandemic dynamics, extending the effect beyond the end of the special period. We can also see that models excel in different parts of the time axis. For example, while ARMean(1) appeared to be overall decent in Figure 2, it produces terrible forecasts in the more recent time spectrum when we consider uncertainty estimation for *Net Income* while maintaining excellent performance on *Total Equity*. There seems to be a visibly recurring pattern in predictability in *Cash from Operations* every four quarters, where the last quarter of each year is the least predictable.

Figure 6 shows the error at each of the four next quarters being predicted for all metrics. Interestingly, the error does not increase arbitrarily at further steps ahead but instead falls again at four quarters into the future. This is likely explained by the unique role of end-of-year reports, which makes them more predictable.

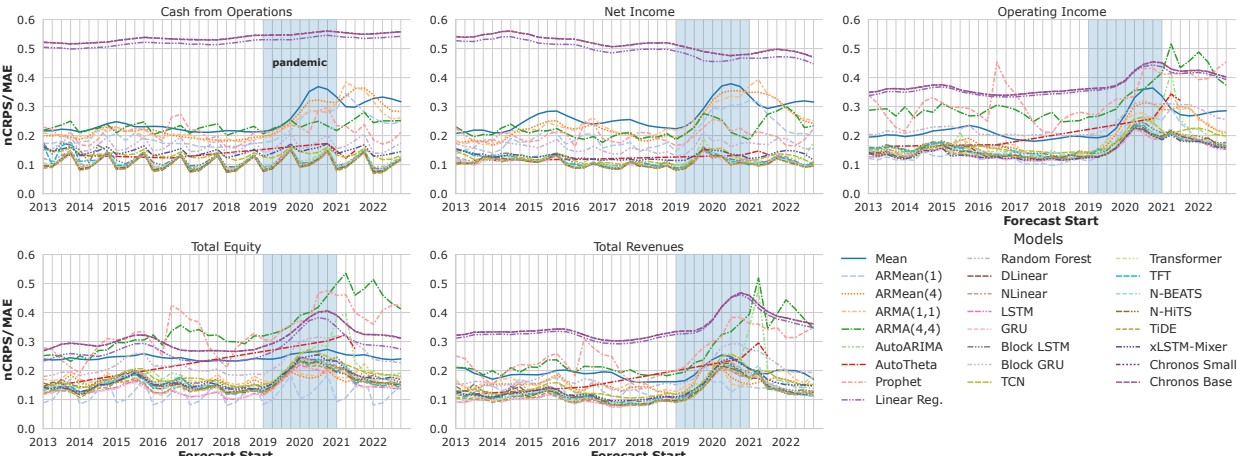

Figure 5: **Periods in which CFs are hard to predict vary across models and features.** We can immediately observe deeply impactful events like the global COVID-19 pandemic manifesting in more erroneous forecasts, as exemplified by the areas highlighted in blue. The mean is taken over all four-quarter forecasts starting at each point.

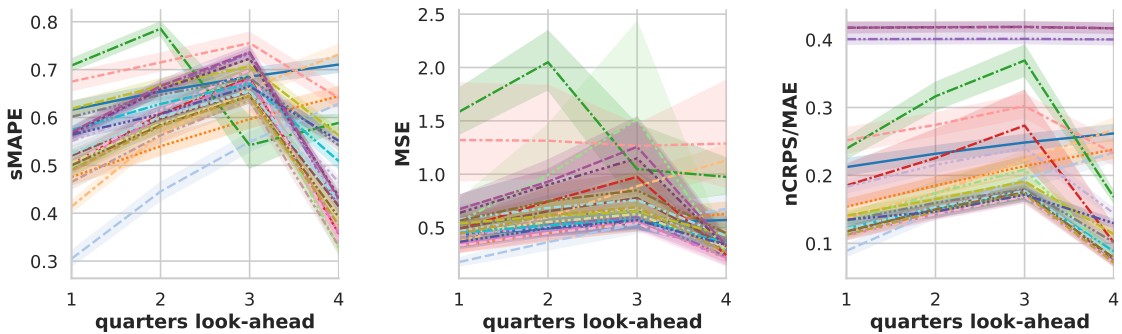

Figure 6: **Forecast error generally rises with increasing look-ahead horizon, with a noticeable drop at four steps.** The error is over all features and folds, where lower is better. The shaded area indicates one standard deviation. See Figure 5 for the legend.

**Comparison to Human Forecasts** We also compared the machine forecasts with those of human expert analysts. To this end, we used data from *StarMine* by *Refinitiv*, which contains one-year-ahead non-probabilistic forecasts of the expected *Total Revenues* for 87.96% of the companies from 2013 Q3 to 2022 Q2. Thus, we need to compare them only with the automatic forecasts for that feature, those companies, the last of the four look-ahead steps, and the valid time window. The results are provided in complete form in Appendix A.6. In summary, automatic forecasts are superior for most metrics, even though different models than in Table 2 are leading. However, human analysts make mispredictions that better match the scale of the data, e.g., minor errors on small scales and large ones on bigger ground-truth numbers. This results in much better MAPE and sMAPE scores and might hint at potentially more calibrated uncertainty estimates if they were available. Given that human forecasts are commonly employed in investing (Guerard et al., 2015), it is interesting to see how their utility compares to automatic forecasts.

## 4.2 Realisitc Market Evaluation

To investigate if the developed high-quality predictions translate to improved investment performance, we apply them to factor models that are backtested in real-world market situations, similar to the LFMs of Alberg & Lipton (2018). For brevity, we investigate GRU as the best model identified in Section 4.1.

Table 3: **Performance of different investment strategies** employing ground truth forecasts (*Clairvoyant*), human analyst expectations, and forecasts from GRU models. We provide the final portfolio value, the compound annual growth rate (CAGR), the annualized volatility, and the Beta. Overall, portfolios based on idealized Clairvoyant and analyst expectations outperform the reference index, as well as one based on the Operating Income forecasts using GRU and rebalanced every twelve months.

|  | Rebalancing | Forecast | Perf. (%, ↑) | CAGR (%, ↑) | Vola. (%, ↓) | Beta (↓) |
|---|---|---|---|---|---|---|
| MSCI World Reference | n.a. | n.a. | 247.25 | 14.69 | 12.50 | 1.00 |
| Operating Income per Enterprise Value | 3 Months | Clairvoyant | 315.42 | 16.97 | 15.41 | 1.13 |
|  |  | GRU | 233.22 | 14.17 | 15.81 | 1.16 |
|  | 12 Months | Clairvoyant | 287.81 | 16.09 | 14.85 | 1.07 |
|  |  | GRU | 257.51 | 15.06 | 15.39 | 1.13 |
| Total Revenues per Enterprise Value | 3 Months | Clairvoyant | 272.74 | 15.59 | 16.89 | 1.24 |
|  |  | Analyst | 312.13 | 16.87 | 15.81 | 1.17 |
|  |  | GRU | 238.48 | 14.37 | 16.27 | 1.19 |
|  | 12 Months | Clairvoyant | 278.89 | 15.80 | 16.47 | 1.20 |
|  |  | Analyst | 326.29 | 17.31 | 15.55 | 1.14 |
|  |  | GRU | 212.76 | 13.38 | 16.23 | 1.19 |

**Experimental Design**  We built simulated portfolios from the CF forecasts based on several different strategies. First of all, we considered two different proxies to identify good stocks, namely, the highest Operating Income or the highest Total Revenues for the four quarters in the future, each normalized by the current Enterprise Value of the company. Secondly, we rebalanced our portfolio either after each quarter or at the start of each new year. In addition to the criteria in Section 4.1.2, we excluded the real estate and financial sectors due to special reporting semantics. To ensure our method is not merely exploiting some current sector-specific performances, we ensure our allocation replicates the sector composition of the *MSCI World* reference index. We similarly approximate its regional distribution as closely as possible. Our portfolios each consist of 50 stocks with a fixed 2% of the capital each. At each rebalancing step, the best possible companies based on the above criteria are chosen, and stocks are sold and bought to reflect that reevaluation. We compare our method to the reference index and two other investment strategies: First, we replicate the Clairvoyant portfolios based on the same selection criteria but using the usually unknowable exact ground truth future (Alberg & Lipton, 2018). Secondly, we compare portfolios using our forecasts with ones based on available human analysts' expectations for total revenues.

**Findings**  The performance of portfolios based on CF forecasts and their baselines is shown in Table 3. We can firstly see that for Operating Income, rebalancing each year is superior to each quarter, and vice versa for Total Revenues, possibly due to the special annual cycle of financial reporting also visible in Figure 6. Secondly, we can confirm that the hypothetical Clairvoyant models are effective tools for portfolio optimization, further motivating the pursuit of improving CF forecasting. Thirdly, human analysts' predictions are highly effective for investing, possibly reinforced through self-fulfilling prophecies and feedback loops (Sant & Zaman, 1996; Mauboussin, 2002). This confirms the empirical findings of Guerard et al. (2015). Lastly, portfolios based on automated GRU forecasts are useful for investing, enabling a final portfolio value that is 10 percentage points larger than the reference index after about ten years when using the Operating Income and rebalancing yearly. While the volatility and Beta are expectedly higher than the reference index, they are on par with the other strategies. Detailed inspection of the entire time span reveals that our method consistently finds better stocks than when following the reference index in the described setting, except for the onset of the pandemic effects in Q2 2020, through to the end of 2020. Similarly, the increased volatility originates to a large degree from the time of 2019 onwards, and was comparably low for the previous years. This, again, matches with the findings in Figure 5, where pandemic uncertainty leaks back into the forecasts starting 2019 Q2. This strongly motivates going beyond the probabilistic forecasting performed so far and investigating methods to maintain reasonable performance in such unstable times, too.

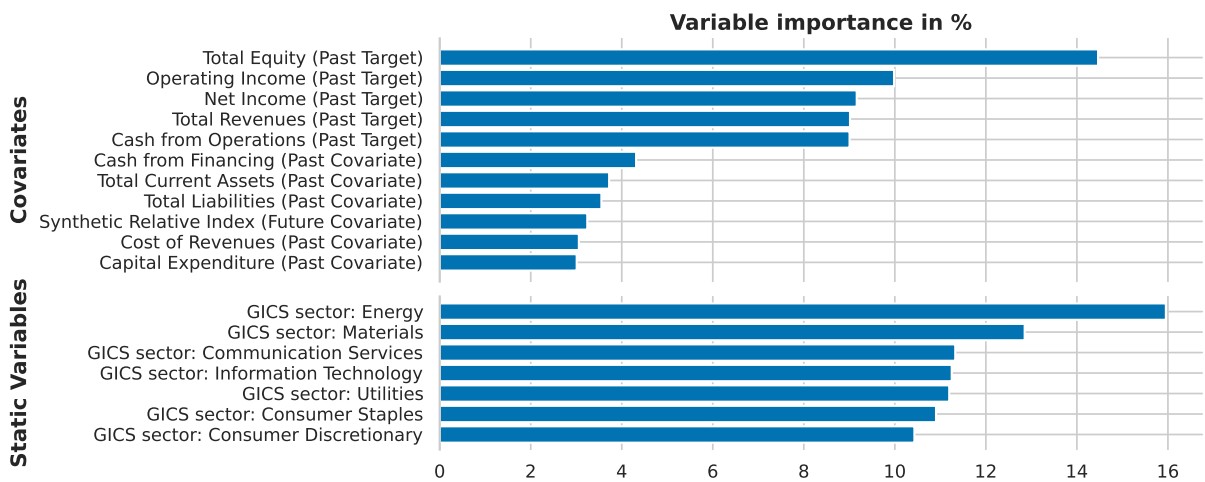

Figure 7: **Only some of the features are relevant for certain forecasts.** The figure shows the exemplary importance of the input variables when forecasting using the TFT model. It shows the probabilistic forecast for *Pacira BioSciences, Inc.* (see Figure 3) over all 40 folds.

## 5 Interpretability, Explanations and Domain Expertise

When building models, it is most often worthwhile to leverage the expertise of practitioners regularly working with the data at hand (Collopy et al., 2001; Kelly & Xiu, 2023, Sec. 1.5). We thus provide an overview of which of the investigated models can reasonably benefit from such information. Most models support simple inductive biases by fixing, bounding, or regularizing certain parameters. However, this is only useful in practice where such weights can be identified with a specific piece of domain knowledge and, therefore, not helpful in most deep architectures. Therefore, it might be necessary to fall back to more general methods of incorporating human insights like explainable interactive learning (XIL), where experts improve the model by providing feedback based on explanations of model outputs (Teso & Kersting, 2019; Kraus et al., 2024c). We will discuss the possibilities of applying interpretability methods to the models at hand to permit such approaches in the first place. This furthermore allows some degree of validation of the modeling outcomes and is therefore highly desirable in most practical ML applications (Collopy et al., 2001; Hoang & Wiegratz, 2023; Steinmann et al., 2024). Also, parameters such as the lookback window or forecast horizon are always opportunities to use expert knowledge and were already discussed in Section 4.1.3. In this section, we thus focus on higher-level and human-understandable aspects specific to each model.

**Local Models**  The simplistic ARMean($p$) model only permits setting the lookback window $p$, thus limiting its overall adaptability. Incorporating domain expertise is easily possible in the case of ARMA($p, q$), ARIMA($p, d, q$), and AutoARIMA models. For instance, we could assume that long-term means likely don't carry much information over shorter ones and therefore set $p > q$, effectively constraining $\psi_i = 0$ for all $i > q$. Seen the other way around, this means that if the model automatically determined such a value (either by parameter estimation or by the search in AutoARIMA), it possibly teaches us something about the dynamics at play. For example, a model mostly focusing on values from four quarters in the past hints at data of yearly seasonality. Alternatively, if it has already been determined that some component is stationary, we could set $d = 0$. Such an opportunity is less obvious in the case of the Theta method, though its relationship to the exponential smoothing with drift model (SES-d) could be exploited for that (Hyndman & Billah, 2003). The Prophet model was specifically designed to allow for human-in-the-loop modeling (Taylor & Letham, 2018). Time series are additively decomposed into trend, seasonality, holiday, and residual components. Each is modeled separately and can accommodate expert insights. For instance, one can specify the type of trend, manually define changepoints at which a linear trend changes, or provide capacities for trends with saturating growth. The seasonality component allows multiple recurrence periods, and the holiday dates can be set according to regional customs. Note, however, that the latter two do not fit the quarterly and international data at hand and, therefore, have little applicability to CF forecasting.

**Global Models**  While Linear Regression permits constraining specific weights, this is hardly desirable, as this model is seldom used as more than a baseline. For most other global models, opportunities for expert involvement are limited. However, general black-box explainability methods like SHAP or Integrated Gradients apply to all deep learning methods (Marcinkevičs & Vogt, 2023), although the exact insight gained from them can be challenging to understand for domain experts. Some of the models provide specific white-box methods to gain insights into the forecasts. For example, models including and building on Transformers (like TFT) allow inspecting the attention map to learn what the model "looks at" (Vaswani et al., 2017) and when attention maps are atypical (Lim et al., 2021). Furthermore, TFT allows inspecting the importance of the input variables. This is shown exemplarily in Figure 7 for all covariates with at least 3%, and static variables with at least 10% importance, respectively. The stacked N-BEATS architecture allows for modeling different task-specific basis functions in different stacks and inspecting the backcasting results. Thus, one can define detrending and seasonality-removing blocks and then inspect and validate their behavior. The same is possible for the hierarchical extension N-HiTS. In summary, all of these methods allow for interpreting the decisions and explaining the output of deep learning models. To eventually improve them, human-in-the-loop methods like XIL can be used, helping to build better and more reliable forecasters.

## 6 Conclusion

We established the need for an in-depth comparison of company fundamentals forecasting models in both qualitative and quantitative terms. This is relevant not only for finance but also as a case study for machine learning and econometrics. We started with an in-depth analysis of the data and solutions to its specific challenges. We explored the applicability of 24 classical statistical and modern machine learning models and identified their properties and capabilities. Subsequently, we have shown that identifying the best overall model requires a nuanced analysis. Particularly, we demonstrated that surpassing simple baselines in deterministic forecasts is challenging. Yet, the much more helpful probabilistic forecasts can be obtained by either AutoARIMA or, equivalently, by any of the deep learning models. These global models can learn from the time series of many companies, effectively estimating a more adequate distribution. We have shown that enriching the data with static context provides negligible benefits, suggesting that the key predictive patterns are already latent in the covariate and target variables. Depending on the field of application and the variables of interest, vastly different models should be selected. For example, a varying set of models excelled before, during, and after the COVID-19 pandemic, respectively. We further find the quality of the obtained forecasts to be similar to human analyst predictions. We continue by evaluating the best automated forecasts on factor models to assess their practical use in investing. We find that they can outperform the reference index over ten years and observe the benefit to be greatest in times of higher financial stability. We finally motivated and showed possibilities for interpreting the models' behavior and generating explanations for their forecasts. This, in turn, permits domain experts to elevate and validate the quality of predictions and eventually improve them.

The greatest challenge to CF forecasting seems to be the large variability of patterns across the years. This is amplified by the relatively small amount of data available. Data upsampling techniques might permit more fine-grained forecasts once higher-frequency data of at least some companies becomes available, from which interpolations might be learnable (Ghosh et al., 2021). The data scarcity and the inherent out-of-sample task makes ensemble methods and incorporating additional signals and domain knowledge appear to be very fruitful. For instance, one could mine textual features from news, investor calls, or social media by extracting sentiment scores or semantic information. Furthermore, causal models might help distinguish spurious correlations, possibly from data collection artifacts, from actual causal relations, like the onset of disruptive global events. This is closely tied to mining patterns in CF data for insights into the macroeconomic behavior of entire sectors or countries in relation to each other. The other way around—zooming in on specific companies—might reveal particularities about individual companies or their relation to other market participants. Company fundamentals forecasting is, furthermore, inherently continual since a new set of fundamentals is released each quarter. Explicit lifelong learning methods might aid in regulating which of the learned patterns still apply in the new time period.

**Aknowlwegements**

This research project was funded by the ACATIS Investment KVG mbH project "Temporal Machine Learning for Long-Term Value Investing" and the German Federal Ministry of Education and Research (BMBF) project KompAKI within the "The Future of Value Creation – Research on Production, Services and Work" program (funding number 02L19C150) managed by the Project Management Agency Karlsruhe (PTKA). The authors of the Eindhoven University of Technology received support from their Department of Mathematics and Computer Science and the Eindhoven Artificial Intelligence Systems Institute. The authors are responsible for the content of this publication.

Additionally, we thank the anonymous reviewers whose comments helped improve this manuscript.

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

# A    Appendix

This supplement provides a discussion of wider ramifications of this work (Appendix A.1), detailed statistics on the CF dataset, highlighting the challenges of the forecasting task (Appendix A.2), details for reproducing the model training (Appendix A.3), insights on the effect of RevIn (Appendix A.4) and providing static context (Appendix A.4), as well as the comparison of human analyst's to the model's forecasts (Appendix A.6).

## A.1    Broader Impact Statement

This work employs publicly known models for widely discussed data where little rigorous evaluation has been presented so far. We, therefore, anticipate minimal potential for harm. However, allocating large quantities of assets solely based on CF forecasts could have adverse macroeconomic effects due to the one-dimensional allocation rule, which overlooks crucial aspects of businesses, such as environmental and societal impacts. Despite this, such a scenario is highly unlikely considering the current diverse landscape of investors. Similarly, future work could compare the quality of forecasts and resulting stock selections across different regions of the world.

## A.2    Charactersitics of the Dataset

Even after the normalization described in Section 4.1.2, the data exhibits significant skewness and heavy tails, effectively making it considerably non-Gaussian. That is shown in Figure 8 and Table 4. This reoccurring problem in finance (Schmitt et al., 2013) invalidates some model assumptions, like of the ARIMA-family models. This might explain some of the observed challenges with forecasting the CFs with those models.

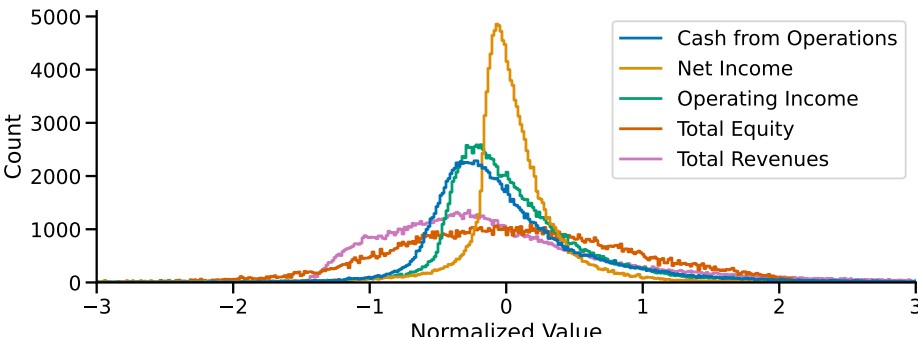

Figure 8: **The five target features are substantially non-Gaussian even after normalization.**

Table 4: **Statistics of the normalized target features.** They are highly non-Gaussian.

| Feature | min | median | max | mean | std | skew | kurtosis |
|---|---|---|---|---|---|---|---|
| Cash from Operations | -64.011 | -0.111 | 105.825 | 0.000 | 1.000 | 5.921 | 1350.920 |
| Net Income | -49.323 | 0.007 | 37.237 | 0.000 | 1.000 | -5.693 | 371.476 |
| Operating Income | -60.257 | -0.051 | 30.921 | 0.000 | 1.000 | -14.990 | 534.002 |
| Total Equity | -17.434 | 0.020 | 13.423 | 0.000 | 1.000 | -2.202 | 25.443 |
| Total Revenues | -1.481 | -0.201 | 17.418 | 0.000 | 1.000 | 2.254 | 12.264 |

A further challenge of the CF dataset is the lack of stationarity. To quantify this common challenge in financial datasets, we applied the Augmented Dickey-Fuller (ADF) (Mackinnon, 1994) and Kwiatkowski-Phillips-Schmidt-Shin (KPSS) (Kwiatkowski et al., 1992) tests each with a significance level of 0.05, as implemented by *statsmodels* (Seabold & Perktold, 2010). Of all 50,540 time series (20 variates times 2527 companies), only 12.2% are stationary according to both.

When tested with the auto-correlation function and a significance level of 0.05, about 42.2% of all time series are seasonal. The frequency of the seasonal components varies significantly, as illustrated in Figure 9. This further challenges machine learning forecasting methods.

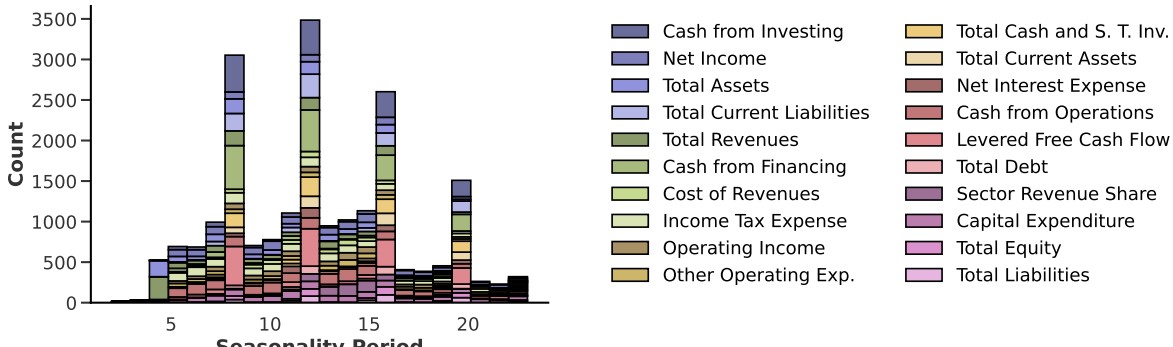

Figure 9: **Very different seasonalities are present in the data, even for the same feature.** Moreover, there is no strong seasonality after four quarters (one year), yet for all other multiples of four. This figure shows a histogram of the strongest seasonalities in each feature.

## A.3 Details on Model Configurations and Training

We based our analysis on the implementation in the *darts* software library (Herzen et al., 2022) and the official implementation of Chronos (Ansari et al., 2024)[1]. We make our implementation openly available[2]. We used *scikit-learn* (Pedregosa et al., 2011) for data normalization. For evaluation, we used *torchmetrics* (Nicki Skafte Detlefsen et al., 2022) and the CRPS implementation of *Pyro* (Bingham et al., 2019). Some models obtain probabilistic forecasts via specialized architectures or sampling methods, while others simply learn to predict the parameters of a distribution instead of the data directly. This is the case for Linear Regression as well as all deep learning models, where we, therefore, performed quantile regression on $0.01, 0.05, 0.1, 0.15, 0.2, 0.25, 0.3, 0.4, 0.5, 0.6, 0.7, 0.75, 0.8, 0.85, 0.9, 0.95, 0.99$.

We introduced instance normalization to obtain a zero mean over the time steps per variate for the Prophet model. This was empirically ineffective for all other local models. For $ARMA(1, 1)$, $ARMA(4, 4)$, and $VARIMA(4, 0, 4)$, we estimated an affine trend. Differencing already eliminated constant trends in $ARIMA(4, 1, 4)$, so a linear trend sufficed.

For the linear regression and random forest models, we estimated separate models for each target variable and future time step. For the latter, each forest contained 100 trees learned from bootstrapped subsamples.

We trained all deep learning models with gradient descent on a batch size of 64 for 100 epochs. We adjusted for different training duration requirements by early stopping after no improvement in the validation loss after three epochs. We employed the AdamW optimizer (Loshchilov & Hutter, 2018) with learning rate $10^{-4}$, weight decay weighting of $10^{-2}$, $\beta_1 = 0.9$, and $\beta_2 = 0.999$. Furthermore, we clipped gradients to 1.0 to stabilize training. For all models except DLinear and NLinear, we used a 10% dropout rate. The following lists the specific hyperparameters for each of the deep learning architectures. For DLinear, we used a kernel size of 10 to estimate the moving average of the trend. All recurrent neural networks (GRU, LSTM, and block variants) were three layers deep. The single models and block variants had a hidden size of 64 and 128, respectively. The TCN convolutions were set to a kernel width of 3 steps, 16 filters, and a dilation of 2 time steps. The Transformer was trained with a token size of 120, a feedforward dimension of 512, GELU activations, four encoder layers, four decoder layers, and six attention heads. TFT used a hidden size of 36 and six attention heads covering the entire time span. We learned a single block of six layers with hidden dimension 512 for N-BEATS. The N-HiTS models consisted of three stacks, each with a single two-layer block and 512 hidden dimensions. Our configuration of TiDE had two encoder and decoder layers each, a hidden size of 128, and a decoder with a hidden size of 32 and an output dimension of 16.

---

[1]https://github.com/amazon-science/chronos-forecasting
[2]https://github.com/felixdivo/Forecasting-Company-Fundamentals

### A.4 On the Use of Reversible Instance Norm

Table 5 shows that using RevIN is beneficial in many but not all models. In the case of DLinear and NLinear, the detrending and normalization operations seem to be already sufficient and render RevIN unnecessary.

Table 5: **RevIN is beneficial for only some of the models.** The scores are the nCRPS obtained from probabilistic forecasts, where lower is better.

| Models | Without RevIN | With RevIN | Improvement |
|---|---|---|---|
| DLinear | 0.125±0.02 | 0.131±0.02 | -4.80% |
| NLinear | 0.127±0.02 | 0.135±0.02 | -6.30% |
| TCN | 0.163±0.03 | 0.153±0.02 | 6.13% |
| Transformer | 0.136±0.02 | 0.132±0.02 | 2.94% |
| TFT | 0.133±0.02 | 0.134±0.02 | -0.75% |
| N-BEATS | 0.134±0.02 | 0.134±0.02 | 0.00% |
| N-HiTS | 0.129±0.02 | 0.130±0.02 | -0.78% |
| TiDE | 0.122±0.02 | 0.126±0.02 | -3.28% |

### A.5 On the Use of Static Features

As shown in Table 1, some models can leverage numerical or categorical context in addition to the plain time series. As laid out in Section 4.1.1, we added GICS sector classifications to each company. In this section, we investigate the impact of this additional information on forecast accuracy. As Table 6 shows, the overall effect of this additional data is marginal. While it slightly improved the performance of NLinear, it worsened the performance of TFT, conceivably due to the larger tendency to overfit.

Table 6: **Most models perform similarly with and without incorporating static features.** This table compares supplying the models with only the time series to also providing static context per company. The scores are the nCRPS obtained from probabilistic forecasts, where lower is better.

| Models | Without Statics | With Statics | Improvement |
|---|---|---|---|
| Linear Reg. | 0.401±0.02 | 0.401±0.02 | 0.00% |
| DLinear | 0.131±0.02 | 0.131±0.02 | 0.00% |
| NLinear | 0.136±0.02 | 0.135±0.02 | 0.74% |
| TFT | 0.132±0.02 | 0.134±0.02 | -1.52% |
| TiDE | 0.126±0.02 | 0.126±0.02 | 0.00% |

## A.6 Algorithmic and Human Forecasts

To facilitate a relation of the human evaluation to the automated forecast, we provide comparable results in Table 7, as discussed in Section 4.1.3. Here, we made sure to test the models only on the companies where expert forecasts were available. Furthermore, only the fourth (one-year-ahead) time step is considered, which aligns with the human expectations. For simplicity, we show the results for an entire company in the model evaluation, even if individual human analyst predictions are missing, since this happens only rarely. Since the expectations contained some heavy outliers, we omitted metric results larger than four standard deviations from the statistics. The comparison is performed on the transformed data as laid out in Section 4.1.2. Consistent with the prior evaluation, the provided nCRPS score for the deterministic analyst expectations is the special case of the MAE. Results are shown for the probabilistic models.

Table 7: **Comparing automatic and human forecasts.** This table is analogous to Table 2, but the evaluation was restricted to the appropriate companies, feature (*Total Revenues*), horizon (4 quarters in the future), and time steps.

| Models | MAE ($\downarrow$) | MSE ($\downarrow$) | RMSE ($\downarrow$) | MAPE ($\downarrow$) | RSE ($\downarrow$) | sMAPE ($\downarrow$) | $R^2$ ($\uparrow$) | nCRPS ($\downarrow$) |
|---|---|---|---|---|---|---|---|---|
| Human Analyst | 0.199±0.03 | 0.273±0.06 | 0.520±0.06 | **0.710±0.30** | 0.230±0.05 | **0.305±0.04** | 0.770±0.05 | 0.199±0.03 |
| Mean | 0.206±0.03 | 0.112±0.03 | 0.332±0.04 | 2.416±3.40 | 0.131±0.04 | 0.520±0.05 | 0.869±0.04 | 0.206±0.03 |
| ARMean(1) | 0.177±0.04 | 0.090±0.03 | 0.296±0.05 | 2.312±2.57 | 0.106±0.05 | 0.464±0.07 | 0.894±0.05 | 0.177±0.04 |
| ARMean(4) | 0.179±0.04 | 0.090±0.03 | 0.295±0.05 | 2.264±2.45 | 0.106±0.05 | 0.470±0.07 | 0.894±0.05 | 0.179±0.04 |
| ARMA(1,1) | 0.244±0.04 | 0.155±0.05 | 0.390±0.06 | 2.709±2.60 | 0.182±0.07 | 0.569±0.07 | 0.818±0.07 | 0.244±0.04 |
| ARMA(4,4) | 0.289±0.05 | 0.229±0.08 | 0.472±0.08 | 3.005±2.44 | 0.268±0.11 | 0.626±0.07 | 0.732±0.11 | 0.289±0.05 |
| AutoARIMA | 0.193±0.04 | 0.126±0.08 | 0.343±0.09 | 2.196±2.57 | 0.148±0.11 | 0.529±0.07 | 0.852±0.11 | 0.193±0.04 |
| AutoTheta | 0.215±0.04 | 0.128±0.04 | 0.354±0.06 | 2.645±2.73 | 0.152±0.07 | 0.521±0.08 | 0.848±0.07 | 0.215±0.04 |
| Prophet | 0.254±0.05 | 0.176±0.06 | 0.415±0.07 | 2.958±3.04 | 0.209±0.10 | 0.580±0.08 | 0.791±0.10 | 0.254±0.05 |
| Linear Reg. | 0.178±0.03 | **0.083±0.03** | 0.284±0.05 | 2.127±2.56 | **0.096±0.04** | 0.468±0.07 | **0.904±0.04** | 0.178±0.03 |
| Random Forest | 0.192±0.03 | 0.092±0.03 | 0.300±0.05 | 2.481±2.85 | 0.107±0.04 | 0.491±0.06 | 0.893±0.04 | 0.192±0.03 |
| DLinear | 0.177±0.03 | 0.088±0.03 | 0.292±0.05 | 2.164±2.83 | 0.103±0.04 | 0.466±0.06 | 0.897±0.04 | 0.177±0.03 |
| NLinear | **0.172±0.03** | **0.083±0.03** | 0.284±0.05 | 2.217±2.92 | 0.097±0.04 | 0.458±0.06 | 0.903±0.04 | 0.172±0.03 |
| LSTM | 0.184±0.03 | 0.090±0.03 | 0.296±0.04 | 2.392±3.07 | 0.104±0.04 | 0.477±0.07 | 0.896±0.04 | 0.184±0.03 |
| GRU | 0.182±0.03 | 0.088±0.03 | 0.293±0.05 | 2.008±1.85 | 0.102±0.04 | 0.473±0.06 | 0.898±0.04 | 0.182±0.03 |
| Block LSTM | 0.192±0.02 | 0.098±0.03 | 0.310±0.04 | 2.425±3.16 | 0.113±0.03 | 0.492±0.05 | 0.887±0.03 | 0.192±0.02 |
| Block GRU | 0.179±0.03 | 0.087±0.03 | 0.291±0.05 | 2.150±2.74 | 0.100±0.04 | 0.471±0.06 | 0.900±0.04 | 0.179±0.03 |
| TCN | 0.193±0.02 | 0.099±0.03 | 0.312±0.04 | 2.473±3.66 | 0.114±0.04 | 0.495±0.05 | 0.886±0.04 | 0.193±0.02 |
| Transformer | 0.177±0.03 | 0.085±0.03 | 0.288±0.05 | 2.366±3.05 | 0.099±0.04 | 0.468±0.06 | 0.901±0.04 | 0.177±0.03 |
| TFT | 0.192±0.02 | 0.099±0.03 | 0.311±0.04 | 2.398±3.45 | 0.114±0.03 | 0.493±0.05 | 0.886±0.03 | 0.192±0.02 |
| N-BEATS | 0.182±0.03 | 0.089±0.03 | 0.295±0.05 | 2.304±2.69 | 0.103±0.04 | 0.477±0.06 | 0.897±0.04 | 0.182±0.03 |
| N-HiTS | 0.177±0.03 | 0.085±0.03 | 0.287±0.05 | 2.162±2.44 | 0.098±0.04 | 0.468±0.06 | 0.902±0.04 | 0.177±0.03 |
| TiDE | 0.179±0.03 | 0.086±0.03 | 0.290±0.05 | 2.254±2.96 | 0.100±0.04 | 0.472±0.06 | 0.900±0.04 | 0.179±0.03 |
| xLSTM-Mixer | 0.200±0.03 | 0.383±0.14 | **0.225±0.03** | 3.045±1.27 | 7528.145 | 0.422±0.02 | -27315.441 | **0.141±0.02** |
| Chronos Small | 0.217±0.03 | 0.649±0.40 | 0.255±0.04 | 3.442±1.51 | 7968.601 | 0.421±0.02 | -27482.442 | 0.409±0.02 |
| Chronos Base | 0.224±0.03 | 0.686±0.34 | 0.264±0.04 | 3.425±1.28 | 8627.039 | 0.423±0.02 | -27558.638 | 0.409±0.02 |

