# OpenReview forum: "Forecasting Company Fundamentals"
_TMLR — Accepted by TMLR_

### Review · Reviewer_JfZR · 2024-11-21

**Summary Of Contributions:**

This paper evaluates a wide range of deep learning and machine learning models for forecasting company fundamentals. The paper presents the superior performance in deep learning models compared with classical models and shows how improved forecasting can enhance automated stock allocation tasks.

**Audience:**

Yes

**Broader Impact Concerns:**

No concerns about the broader impact.

**Claims And Evidence:**

Yes

**Requested Changes:**

1.	Explain the “highly atypical behavior” and provide the models’ results on these behaviors.
2.	Include other modalities if possible.
3.	Include advanced learning techniques on the investigated models.

**Strengths And Weaknesses:**

Pros:
+ This paper presents a comprehensive evaluation of a diverse set of machine learning techniques and a wide range of indicators for predicting company fundamentals. The experimental design is well thought out.
+ The paper compares machine learning-based forecasts with human expert analysts and shows the superior performance of machine learning-based forecasts. This finding is insightful.
+ The capabilities of interpretability, explanations, and incorporation are discussed in the proposal, which is innovative.
+ The paper is well-written and easy to follow.

Cons:
- The dataset removes 10 companies due to their “highly atypical behavior,” but the criterion for defining such behavior is unclear. It would also be interesting to see the performance of the machine-learning models in these companies.
- In the real world, the company fundamentals are very likely to be influenced by factors that are not reflected in public data. It is unclear how the proposed evaluation process tackles this issue. In addition, it would be great if the paper could incorporate data from other sources and modalities, such as news and social media, and present the performance of some multi-modality models.
- The evaluation is conducted upon standard machine learning pipelines. It would be interesting to see the model’s performance when integrated with advanced techniques such as self-supervised learning and reinforcement learning.

---

> ### Author Response · Authors · 2025-02-04
> **Individual Response to JfZR**
>
> Dear Reviewer,
>
> Thank you so much for your constructive review, appreciating the thorough analysis, comparison with human experts, and discussion of interpretability.
>
> **1. Regarding the “highly atypical behavior”:** We tried to remove as few companies as possible to still cover the diverse range of companies. After applying the data normalization described in Sec. 4.1.2 on all companies, we identified and removed those with more than one data point in any feature that was at least $30\sigma$ from the mean. We found two rounds of such cleaning of extreme outliers to be sufficient. Note that the data is significantly non-Gaussian (cf. App. A.2), which explains that there are actually companies in the $30\sigma$ region. We clarified this in the manuscript (see the highlighted passage in Sec. 4.1.2).
>
> We furthermore also evaluated the performance of all trained models over all time spans on the the outliers we otherwise removed. The results are presented in the following table. The performance is decent for many models. However, nine other models, both local and global, diverged completely while producing a forecast. This is indicated by "—" in the table below. We, therefore, excluded them from the main comparison.
>
> | Models        | nCRPS/MAE   |
> |:--------------|:------------|
> | Mean          | —           |
> | ARMean(1)     | —           |
> | ARMean(4)     | —           |
> | ARMA(1,1)     | 0.195±0.09  |
> | ARMA(4,4)     | 0.276±0.13  |
> | AutoARIMA     | 0.137±0.07  |
> | AutoTheta     | 0.145±0.08  |
> | Prophet       | 0.254±0.15  |
> | ------------- | |
> | Linear Reg.   | —           |
> | Random Forest | —           |
> | ------------- | |
> | DLinear       | —           |
> | NLinear       | —           |
> | LSTM          | 0.117±0.07  |
> | GRU           | 0.119±0.07  |
> | Block LSTM    | 0.185±0.13  |
> | Block GRU     | 0.132±0.10  |
> | TCN           | 0.149±0.08  |
> | Transformer   | 0.131±0.12  |
> | TFT           | —           |
> | N-BEATS       | 0.130±0.09  |
> | N-HiTS        | 0.142±0.11  |
> | TiDE          | —           |
> | ------------- | |
> | Chronos Small | 0.235±0.08  |
> | Chronos Base  | 0.235±0.08  |
>
> **2. Regarding incorporating more modalities:** We agree that there is definitely more useful context to company fundamentals, some of which resides in other modalities. Leveraging it for yet more accurate or trustworthy forecasts is, however, beyond the scope of this work, in which we provide a thorough overview of many forecasting methods for company fundamentals for the first time. As one of the key steps forward, we discuss it in the paragraph on future work in Sec. 6.
>
> **3. Regarding advanced learning techniques:** We gladly take the inspiration to assess how advanced learning paradigms fare at forecasting company fundamentals. Specifically, we identified pretrained time series forecasting models as inspired by natural language or vision-language foundation models. Chronos (Ansari et al., 2024) is a state-of-the-art pretrained forecaster that can be used on CFs. We, therefore, conducted the necessary experiments, amended the manuscript in the respective places, and highlighted the changes in Tab. 1, Sec. 3.2, Sec. 4.1.3 including the various figures and tables, and App. A.5.  We also added sections to Fig. 1 & 2 for improved overview. We generally find Chronos to provide inferior performance, including some serious outliers. This might be caused by the unusual shortness of the CF time series compared to what the model has been pretrained on. Overall, this incorporation of cutting-edge methods rounded off the comprehensive evaluation.
>
> Thank you again for your suggestions!
>
> The Authors
>
> *References*
> Ansari, A. F., Stella, L., Turkmen, A. C., Zhang, X., Mercado, P., Shen, H., Shchur, O., Rangapuram, S. S., Arango, S. P., Kapoor, S., Zschiegner, J., Maddix, D. C., Wang, H., Mahoney, M. W., Torkkola, K., Wilson, A. G., Bohlke-Schneider, M., & Wang, B. (2024). *Chronos: Learning the Language of Time Series.* Transactions on Machine Learning Research (TMLR). https://openreview.net/forum?id=gerNCVqqtR.

---

### Review · Reviewer_ZfA4 · 2025-01-13

**Summary Of Contributions:**

In this paper the authors try to assess the fundamentals of forecasting companys' financial and overall success stability. They take inspiration from multi-variate time series predictions and conduct a thorough evaluation of 22 deterministic and probabilistic models for forecasting company fundamentals (CFs). It bridges a critical gap in assessing machine learning (ML) and statistical methods for time series forecasting of CFs, covering data normalization, model classification, and backtesting in financial contexts. The study highlights the superior performance of deep learning (DL) models, especially RNN variants, in predicting CFs compared to classical approaches. The research also investigates interpretability and expert integration, underscoring potential improvements through hybrid approaches. The findings provide actionable insights for investment strategies and future work in macroeconomic modeling.

**Audience:**

Yes

**Broader Impact Concerns:**

Could the authors expand on the generalizability of this approach to forecast company fundamentals for niche industries?

**Claims And Evidence:**

No

**Requested Changes:**

1. I recommend authors look into modeling the cyclic features as fixed effects - check this paper for reference https://s3.us-east-1.amazonaws.com/climate-change-ai/papers/icml2021/56/paper.pdf.

**Strengths And Weaknesses:**

>Strengths:

*Comprehensive Evaluation*: The work tests an extensive array of models, ranging from ARIMA to advanced deep learning architectures such as LSTM and Transformers.

*Uncertainty Quantification*: Demonstrates the utility of probabilistic forecasts by comparing performance against deterministic predictions.

*Interpretability Insights*: Provides attention to explainability using SHAP, Integrated Gradients, and interpretable model components.


>Weaknesses:

1. The authors use a quarterly data periodicity and although this aligns with the data available, cyclic trends in companys' financial and overall success stability could be varying at a higher fidelity.  Use of bi-directional LSTM with high fidelity data format for forecasting may lead to better results.
2. Although the authors focus on a particular global event: Covid-19, the
3. Static features in the data (that are not cyclic/change over time) was minimally explored.

---

> ### Author Response · Authors · 2025-02-04
> **Individual Response to ZfA4**
>
> Dear Reviewer,
>
> Thank you so much for your insightful review. We appreciate you recognizing our in-depth analysis employing many different models while considering uncertainty quantification and interpretability.
>
> **1. Regarding the fidelity of the data:** The company fundamentals data we use (see Sec. 4.1.1) is only available quarterly, as compiling the underlying financial reports is a significant effort for large companies of the sizes considered in this work. The same holds for the labor-intensive, partly manual data cleaning performed by the data provider and us. We agree that higher-capacity models, such as bi-directional LSTMs, are worth exploring once data becomes available at a higher cadence. To our understanding, the method of Ghosh et al. (2021) is not directly applicable as of now since we do not have access to time spans of higher than quarterly sampled data. We thus added this idea and the paper of Ghosh et al. (2021), which you recommended as interesting future work, to the manuscript (see the highlighted text in Sec. 6).
>
> **2. Regarding static features in the data:** We agree that there might be further potential in finding static structure in the data, such as company-specific behavior that is constant across time. It is a promising direction for future in-depth analysis after this first-ever survey on company fundamentals. We added a discussion of this to the paper (we again marked the changes to Sec. 6 in the updated PDF).
>
> Thank you again for your suggestions!
>
> The Authors
>
> *References*
>
> Ghosh, R., Craig, M., Matthews, H. S., & Berti-Equille, L. (2021): *Reconstruction of long-term historical electricity demand data.* ICML 2021 Workshop on Tackling Climate Change with Machine Learning. Retrieved from https://www.climatechange.ai/papers/icml2021/56.

---

### Review · Reviewer_qHo2 · 2025-01-25

**Summary Of Contributions:**

This paper provides a comprehensive comparison of classical statistical and modern machine learning models on the task of forecasting company fundamentals. The authors did in-depth analysis to the results, demonstrating the performance difference of statistical methods and machine learning models in different settings, and provided several useful insights. The authors also compared the model’s effectiveness with human forecasts and assessing their the availability of uncertainty estimates.

**Audience:**

Yes

**Claims And Evidence:**

Yes

**Requested Changes:**

NA

**Strengths And Weaknesses:**

Strengths:
- A comprehensive evaluation to the classical methods and machine learning methods on the important topic of forecasting company fundamentals.
- This paper is well-written and has a nice flow of ideas.
- In-depth analysis on the results, which provides insights to the practitioners on selecting the correct methods for forecasting company fundamentals.

Weaknesses:
I am not an expert of ML applications in economics, I am wandering if the authors plan to opensource the code or to build a benchmark for ML methods on forecasting company fundamentals, which could greatly help the ML for economics community.

---

> ### Author Response · Authors · 2025-02-04
> **Individual Response to qHo2**
>
> Dear Reviewer,
>
> Thank you so much for your valuable feedback and for recognizing the work as the first to provide a thorough evaluation of the important topic of company fundamentals forecasting.
>
> **Regarding the implementation and data availability:** The data provider S&P Global, unfortunately, does not permit data redistribution, even in derived form. This is a fundamental issue of large-scale research on economics that many institutions face. We, therefore, had to resort to instead meticulously describing the data features and pre-processing procedures in Sec. 4.1.1-4.1.2 and App. A.2-A.3. As laid out in App. A.3, our implementation mainly wraps around the *darts* software library (Herzen et al., 2022) and other toolkits. We absolutely agree that the availability of high-quality datasets and implementations is critical for machine learning research. However, in light of the unavailability of the data, we do not see a substantial benefit in providing such a dataset-specific implementation where the dataset is -- regrettably -- inaccessible. We clarified that the data is proprietary in the manuscript (see the highlighted text in Sec. 4.1.2). As future work, we will definitely consider ways to push for more accessible data sources. In the long run, economics research using machine learning will strongly benefit from standardized and open benchmarks.
>
> Thank you again for your suggestions!
>
> The Authors

---

### Author Response · Authors · 2025-02-04
**Author Response**

Dear Reviewers,

Thank you for carefully reading our manuscript, recognizing it as a valuable contribution to time series forecasting research, and suggesting improvements to it.

We carefully read them and responded to them individually below. As suggested, we updated the PDF with clarifications and new experiments on pretrained time series models.

We color-coded the changes to make it easy for you to identify them:
- Reviewer *qHo2*: violet
- Reviewer *ZfA4*: brown
- Reviewer *JfZR*: teal

Best

The Authors

---

### Decision · Action_Editor_EfR7 · 2025-03-17

**Recommendation:** Accept with minor revision

**Comment:**

The paper did in-depth analysis of a variety of time series models for forecasting company fundamentals. The experiments are discussed in details and yield some useful insights. However, the analysis results are highly dependent on the dataset, which is not publicly available and the authors does not plan to share with the paper. Minor changes are required in order for the paper ready for acceptance: adding more experiments on static features and sharing the code so that others can reproduce the results.

**Audience:**

The paper would be of interest to some researchers in time series models for finance applications.

**Claims And Evidence:**

The paper aims to evaluate the effectiveness of existing deterministic and probabilistic models for forecasting company fundamentals. The experiments are carefully designed, and the experiment analysis is well supported.

One concern was raised regarding the static features. The authors added discussions but failed to address the concern in a convincing way.

Another concern is about the reproducibility of the experiments since the data is not publicly available.

---

> ### Author Response · Authors · 2025-04-28
> **Revision**
>
> Dear Action Editor EfR7,
>
> We have uploaded a revised version of the paper. In particular:
>
> - We added additional experiments on static features in the newly added Appendix A.5.
> - We share the code used during experimentation, which is now referenced in Appendix A.3 (See the footnote on p. 23).
> - We added xLSTM-Mixer as an additional model because an implementation has since become available for the darts library we employ.
>
> We marked all those changes in green to make it easy for you to verify the changes.
>
> Best
>
> The Authors

---

> ### Author Response · Authors · 2025-05-22
> **Camera-Ready Version Uploaded**
>
> Dear Action Editor,
>
> In accordance with the OpenReview notifications, we have uploaded the camera-ready version of our submission. We would appreciate it if you could kindly confirm that everything is in order.
>
> Best Regards,
>
> The Authors

---

> > ### Comment · Action_Editor_EfR7 · 2025-05-23
> >
> > Yes. This was approved yesterday.